# Kinetics of Chemisorption on the Surface of Nanodispersed SnO_2_–PdO_x_ and Selective Determination of CO and H_2_ in Air

**DOI:** 10.3390/s23073730

**Published:** 2023-04-04

**Authors:** Alexey Vasiliev, Alexey Shaposhnik, Pavel Moskalev, Oleg Kul

**Affiliations:** 1Department of Natural Sciences and Engineering, Dubna State University, 143407 Dubna, Russia; 2Department of Chemistry, Voronezh State Agrarian University, 394087 Voronezh, Russia; 3Department of Applied Mathematics and Mechanics, Voronezh State Technical University, 394006 Voronezh, Russia; 4C-Component, LLC, 125362 Moscow, Russia

**Keywords:** sensitivity, MOX sensor, temperature modulation, kinetics and mechanism of oxidation, qualitative analysis, quantitative analysis

## Abstract

In this work, the kinetics and mechanisms of the interaction of carbon monoxide and hydrogen with the surface of a nanosized SnO_2_–PdO_x_ metal oxide material in air is studied. Non-stationary temperature regimes make it possible to better identify the individual characteristics of target gases and increase the selectivity of the analysis. Recently, chemometric methods (PCA, PLS, ANN, etc.) are often used to interpret multidimensional data obtained in non-stationary temperature regimes, but the analytical solution of kinetic equations can be no less effective. In this regard, we studied the kinetics of the interaction of carbon monoxide and hydrogen with atmospheric oxygen on the surface of SnO_2_–PdO_x_ using semiconductor metal oxide sensors under conditions as close as possible to classical gas analysis. An analysis of the influence of catalytic surface temperature on the mechanisms of chemisorption processes allowed us to correctly interpret and mathematically describe the electrophysical characteristics of the sensor in the selective determination of carbon monoxide and hydrogen under nonstationary temperature conditions. The reaction mechanism is applied as well to the analysis of the operation scheme of the CO sensor TGS 2442 of Figaro Inc.

## 1. Introduction

Many problems of gas analysis are successfully solved by mass spectrometers, chromatographs, and chromato-mass spectrometers; however, these instruments, which combine high sensitivity, selectivity, and stability, are big and expensive. In addition, they need qualified personnel and services. At the same time, there are important problems of gas analysis that can be solved using inexpensive compact gas analytical instruments based on chemical sensors, for example, the problems of continuous environmental monitoring in hard-to-reach places [1,2].

Metal oxide (MOX) sensors used to detect toxic, flammable, and explosive substances in air are characterized by high sensitivity and small size [3,4]. However, the sensitivity and selectivity of MOX sensors are not always sufficient for solving important practical problems. One way to increase the sensitivity and selectivity of MOX sensors is to use nonstationary temperature regimes [5].

The reasons for increasing the sensitivity of the sensor under non-stationary temperature regimes can be associated, for example, with the rapid activation of the catalyst during intense heating. In this case, the catalyst interacts with a significant number of gas-analyte molecules sorbed on the surface of the gas-sensitive layer at the stage of its cooling, which don’t have time to be desorbed at the stage of its intensive heating. The reason for an increase in selectivity of the analysis is the separation in time of the processes of chemisorption and chemical interaction, which reveal the individual features of analyte gases [6].

In most works devoted to non-stationary temperature regimes, the temperature of MOX sensors was modulated by harmonic oscillations with a constant period [5,7,8,9,10,11,12,13,14,15,16,17] or close to it [18]. In other works, researchers used a pseudo-random oscillation generator to modulate the temperature [19,20,21], and in some works, researchers used sharp, “pulse” modes of sensor heating [6,22,23,24,25,26,27,28].

In stationary modes, the sensor response is a scalar quantity, but in non-stationary modes, the sensor response is a vector quantity when one concentration of each of the analyzed gases corresponds to a set of many electrical conductivity values that change over time during the measurement cycle. The nature of the change in electrical conductivity depends on many factors: the analyte gas and its concentration, the gas-sensitive material, and the temperature modulation of the sensor during the measurement cycle. Qualitative and quantitative analyses of the gaseous medium in this case requires mathematical processing of multidimensional data arrays, which is usually carried out on the basis of various methods of dimensionality reduction. Initially, researchers used for this purpose Fourier transforms [4,5,7,8,9,10,15] or wavelet transforms [11,13,14,15], but recently they have begun to use various projection methods [26,27,28] and artificial neural networks [18]. The mechanism of chemisorption and chemical interactions on the surface of metal oxide sensors was considered in publications [29,30,31,32,33].

The mathematical methods that have recently been used to process multidimensional data are suitable for more or less productive computers but are too complicated for simple microcontrollers. Meanwhile, the main task when using chemical gas sensors is to design inexpensive, compact devices that operate autonomously in hard-to-reach places. In the development of such devices, no computer processors are used, but relatively simple microcontrollers with limited speed, memory, and power consumption. Thus, we can formulate the problem of developing methods for processing multidimensional data that would not require the use of significant computing resources.

The least computational resources in processing test data for the electrical conductivity of the MOX sensor on time (temperature) are required, if the analytical dependence corresponding to the data of the training sample is known. Calculating the concentration of the analyzed components, in this case, should not cause problems even when using relatively simple and low-power microcontrollers. However, determining the analytical dependence of the electrical conductivity of a MOX sensor on time (or temperature) can be a very difficult task. The electrical conductivity of a MOX sensor depends on many factors: the temperature changes, both the intrinsic electrical conductivity of the semiconductor and the chemisorption of oxygen and analyte gas change. In addition, the rate of redox reactions between the analyzed gas and chemisorbed oxygen changes. Even in the case of a slow and continuous change in the surface temperature of the gas-sensitive layer, determining the combination of many factors that affect the electrical conductivity of the sensor is an extremely difficult task. However, when analyzing a gaseous medium, temperature regimes of sensor operation with very sharp heating are often used, because these regimes can significantly increase the sensor’s response. In this case, the determination of the analytical dependence of electrical conductivity on time (or temperature) becomes even more complicated since a significant deviation of the system from the equilibrium state can change the mechanisms of these processes. However, the objective difficulty of the task of finding analytical, functional dependencies does not mean that they cannot be found in principle. At least some stages of the measurement cycle (for example, a rather slow cooling) are more or less close to equilibrium processes and can be described by simple equations relating the electrical conductivity of the sensor material with time (temperature) in non-stationary measurement modes. To find an analytical functional relationship between the electrical conductivity of the sensor and its temperature, we need to investigate the mechanisms of chemical and chemisorption processes. This cannot be carried out within the framework of a conventional sensory experiment that studies only the electrical conductivity of the gas-sensitive layer. We will need to investigate the nature of the products of the corresponding chemical interactions as well as the dependence of the yield of products on the temperature of the catalytic surface. To do this, we should carry out the experiments to study the kinetics of heterogeneous catalytic reactions. On the basis of such experimental data, we will be able to plot the functional dependencies of electrical conductivity on time for some sections of the measurement cycle under non-stationary temperature conditions for the MOX sensor.

## 2. Materials and Methods

### 2.1. Sensors Fabrication

To fabricate gas-sensitive layers of metal oxide sensors, we used a highly dispersed powder of tin dioxide obtained from tin acid. To do this, we added dropwise a chilled concentrated ammonia (Sigma-Aldrich Cas No 1.05423, St. Louis, MO, USA) solution to a cooled solution of tin acetate (+4) Sigma-Aldrich Cas No 345172 in glacial acetic acid (Sigma-Aldrich Cas No 64-19-7):Sn(CH_3_COO)_4_ + 4NH_3_ + 3H_2_O→ H_2_SnO_3_↓ + 4CH_3_COONH_4_.(1)

We separated the tin acid by centrifugation, washed it with deionized water, dried it, and calcined it to 773 K, as a result of which we obtained a tin dioxide nanopowder:H_2_SnO_3_→SnO_2_ + H_2_O.(2)

The composition and nanostructure of SnO_2_ powders have been characterized by X-ray diffraction (XRD), transmission electron microscopy (TEM) Jeol JEM-2100 (Tokyo, Japan) [26], X-ray photoelectron spectroscopy (XPS) Thermo ARL X’TRA (Thermo Fisher Scientific, MA USA) and X-ray near-edge absorption spectroscopy (XANES) [34]. In the present study, we used the results obtained in [34], X-ray photoelectron spectroscopy (XPS) and X-ray absorption near edge structure (XANES), both employing the high-brilliance synchrotron radiation of the BESSY II storage ring at the Helmholtz Zentrum Berlin on the joint Russian–German beam line.

In addition to tin dioxide particles, the gas-sensitive layer also contained palladium particles, predominantly in the form of palladium oxide PdO_x_. To do this, we added an aqueous solution of tetraamminepalladium nitrate (+2) Sigma-Aldrich Cas No 13601-08-06 to the tin dioxide powder, dried it, and heated it to 573 K. The resulting powder, consisting of a mixture of tin dioxide and palladium oxide particles, was mixed with a viscous vehicle to obtain a homogeneous paste. We applied the resulting paste as thinly as possible on a dielectric substrate made of aluminum oxide with platinum electrodes and a platinum heater. After that, the substrate was dried at a temperature of 363 K and heated to a temperature of 1023 K, as a result of this procedure a highly dispersed brittle gel was formed from the paste. The palladium contained in the complex compound was converted into a mixture of oxides; its content in the gas-sensitive layer was 3% by weight. The resulting material has been characterized by X-ray diffraction (XRD) and transmission electron microscopy (TEM). As our studies have shown, the grain size of the SnO_2_ nanopowder according to transmission electron microscopy was about 5 nm, and the size of crystallites according to X-ray diffraction was of about 2–3 nm, that is, the grains consist of several crystallites [26], which is confirmed by the results of X-ray photoelectron spectroscopy (XPS) and X-ray near-edge absorption spectroscopy (XANES) [34]. Dielectric substrates with heating and gas-sensitive layers were soldered into a standard TO-8 metal package.

### 2.2. Sensor Measurements

Sensing properties were studied on a setup (“Microgas-F”, company Delta-S, Moscow, Russia) equipped with mass-flow controllers at a constant flow rate of 200 mL/min. Reference gas mixtures of “carbon monoxide-air” and “hydrogen-air” were used as a source of gas; they were diluted with synthetic air, a mixture of purified nitrogen, and purified oxygen (21%). All gases were purchased from a company NIIKM, Moscow, Russia. The required humidity of the gas was achieved by passing part of the airflow in the form of small bubbles through distilled water, and as a result, this part of the airflow was saturated to relative humidity close to 100%.

The metal oxide sensor was placed in a stainless steel test chamber. The sensor temperature was controlled by measuring the resistance of the heating layer (Pt-Al_2_O_3_-glass composite) on the sensor substrate. A specially designed device not only measured the electrical conductivity of the gas-sensitive layer of the sensor at a frequency of 38.3 Hz but also controlled the temperature regime of the sensor.

For qualitative and quantitative analyses of a quasi one-component gas systems (air is considered as single component), the array of experimental data was divided into training and test sets. As shown in Figure 1, a change in sensor temperature leads to a significant change in the electrical conductivity of the gas-sensitive layer. In the first section (2 s), the sensor is heated from 373 K to 723 K. In this section, the electrical conductivity of the gas-sensitive layer increases due to an increase in the concentration of charge carriers in the semiconductor. In the second section (13 s), the sensor cools down from 723 K to 373 K. The general trend is aimed at reducing the electrical conductivity of the gas-sensitive layer. However, when determining hydrogen at the initial stage of sensor cooling, one can observe an additional extremum, which is characteristic of hydrogen. The nature of this extremum is apparently associated with the ability of hydrogen molecules to dissociate into atoms and oxidize to cations that can make an additional contribution to the electrical conductivity of the gas-sensitive layer. When determining carbon monoxide in the cooling section, one can observe a minimum of electrical conductivity, after which its increase follows. This increase in electrical conductivity is a specific feature of carbon monoxide and is associated with a change in the mechanism of CO’s interaction with the palladium surface.

The first 8–12 cycles of measurements in the temperature modulation mode were not used for further processing, since the instrument readings stabilized after 2–3 min of its operation. To characterize each concentration of the studied gas system in a non-stationary temperature regime, 30 to 50 measuring cycles were recorded.

The sensor response *S* was calculated as the ratio of the electrical conductivity of the sensor in the medium under study σ to the electrical conductivity of the sensor in pure synthetic air σ_0_ at 14.5 s after the start of the measurement cycle:*S* = σ(τ = 14.5)/σ_0_(τ = 14.5).(3)

The discretization of electrical conductivity values of the gas-sensitive layer σ during each measurement cycle lasting 15 s was carried out at a frequency of about 38.3 Hz and generated a sample of 575 values. The choice of a time point of 14.5 s from the beginning of the measurement cycle was due to the maximum difference between the values of the electrical conductivity of the sensor obtained at different concentrations of analyte gases.

### 2.3. Kinetics of CO Oxidation on Gas Sensitive Layer

It was important for us to compare the results obtained by measuring the concentration of carbon monoxide using metal oxide sensors operating in a non-stationary mode. This was carried out with the results of the direct measurements of the rate of the chemical reaction of carbon monoxide oxidation on a catalyst containing oxide nanoparticles decorated with palladium. To do this, we used a gas-dynamic setup described in detail in [35].

This experimental setup contained three channels equipped with mass-flow controllers, through which flows of oxygen, nitrogen, and carbon monoxide were supplied to the gas chamber at specified flow rates. The concentration of carbon monoxide at the inlet and outlet of the reactor was measured by an optical device for the absorption of radiation in the IR region. The temperatures of both the furnace and the gas inside the reactor were determined using a thermocouple. The catalyst was deposited on a 22 mm × 2 mm alumina ceramic bar, which was placed along the axis of a tubular reactor placed inside a furnace that provided a uniform, predetermined temperature throughout the reactor volume.

All experiments were carried out under isothermal conditions, without a temperature gradient between the catalyst surface and the gas flow, with a laminar gas flow without diffusion restrictions on the chemical reaction rate on the catalyst. As a result, the rate of diffusion to the catalyst surface from the gas phase was significantly higher than the rate of the chemical reaction on the catalyst surface.

One of the goals of these experiments was to avoid the so-called “ignition” or light-off of the catalyst. It is traditionally believed that with an increase in the temperature of the catalyst, as the rate of the chemical reaction increases, a heat transfer crisis occurs and the temperature of the catalyst begins to grow uncontrollably, forming a positive feedback loop: an increase in temperature—an increase in the reaction rate—an increase in heat release—a further increase in temperature. It is believed that this positive feedback forms the “ignition” of the catalyst. In order to ensure that this process is avoided, we conducted studies at low concentrations of carbon monoxide (not higher than 2000 ppm), when the temperature increase may not exceed several degrees Celsius, that is, almost under isothermal conditions.

All these efforts were directed toward using the data from the kinetic experiment to analyze the results obtained when the metal oxide carbon monoxide sensor was operated in a non-stationary mode.

Three types of experiments were carried out, which allowed us to characterize the kinetics of the process of catalytic oxidation of carbon monoxide on catalysts containing noble metals:Measurement of the temperature dependence of carbon monoxide concentration after the catalyst at a constant carbon monoxide concentration and at a constant gas flow rate at the reactor inlet. These are the so-called catalyst light-off curves. As will be shown below, these curves are not thermal but purely kinetic in nature; however, by tradition, we will keep the traditional name. To analyze the experimental curves, we used the Sigma Plot program;Measurement of the dependence of the concentration of carbon monoxide at the outlet of the reactor on the gas flow rate and, therefore, on the contact time of the gas with the catalyst. In this case, the gas concentration at the reactor inlet and the reactor temperature were maintained constant. For a first-order reaction, graphs plotted in the logarithm of concentration—residence time in the reaction zone (reverse flow rate) coordinates should be straight lines;Measurement of carbon monoxide concentration at the reactor outlet as a function of its concentration at the reactor inlet. In this case, the reactor temperature and flow rate were kept constant. For a first-order reaction, graphs plotted in the coordinate conversion—inlet concentration should be straight lines parallel to the concentration axis.

## 3. Results

### 3.1. Sensor Conductivity in Non-Stationary Temperature Mode

The problem of qualitative and quantitative determinations of an analyte gas in a quasi one-component gas system occurs in cases, where the simultaneous presence of two or more analyte gases is hardly possible. In this case, the problems are how to recognize the analyzed gas (qualitative analysis) and how determine its concentration (quantitative analysis). To solve these problems, the dependences of electrical conductivity on time were obtained over one measurement cycle for carbon monoxide in air (Figure 2) and hydrogen in air (Figure 3) systems for the concentrations from 0 to 100 ppm. Similar data obtained over three cycles of measurements for a gas concentration of 100 ppm are shown in Figure 1. Significant differences in the shape of the curves provide a fundamental opportunity for a qualitative analysis of the selected gas systems.

Figure 2 and Figure 3 show that curve 1 corresponds to a change in the electrical conductivity of the sensor in pure synthetic air during 1 measurement cycle, and curves 2–7 correspond to the change in the electrical conductivity of the sensor in synthetic air with impurities of carbon monoxide (Figure 2) or hydrogen (Figure 3) with successively increasing concentrations: 2, 5, 10, 20, 50, and 100 ppm.

Comparing Figure 2 and Figure 3, we could note that the shapes of the electrical conductivity curves of the SnO_2_–PdO_x_ sensor are significantly different. First of all, this applies to the section of the curves corresponding to cooling (the interval from 5 to 15 s after the start of the measuring cycle). While in the air or in the hydrogen medium the electrical conductivity practically did not change, in the carbon monoxide medium it increased noticeably. Usually, the electrical conductivity of a semiconductor decreases with decreasing temperature; therefore, an increase in electrical conductivity found in the experiment is of both theoretical and practical interest. To study it, we conducted a special experiment.

### 3.2. Kinetics of Carbon Monoxide Oxidation

When studying the process of carbon monoxide oxidation on the surface of a palladium-containing catalyst, it was shown that an abrupt increase in this process is not of a temperature but of a concentration nature. This acceleration is not caused by a heat transfer crisis but by a change in the mechanism of a chemical reaction when a certain threshold concentration of carbon monoxide is reached. Moreover, at a lower concentration of carbon monoxide, the reaction rate can be 4–5 times higher than at a higher concentration of carbon monoxide. In this case, the light-off of the catalyst with increasing temperature is a result of a decrease in the actual concentration of carbon monoxide on its surface resulting from the acceleration of the chemical reaction. That is, positive feedback in the process of the catalyst light-off does not occur due to an increase in temperature (temperature increase—acceleration of a chemical reaction—crisis of heat transfer—a new increase in temperature), but due to a change in the reaction mechanism with a decrease in concentration (temperature increase—acceleration of the reaction—a decrease in the concentration of carbon monoxide—a further acceleration of the reaction). That is, the light-off of the catalyst actually occurs under isothermal conditions.

The empirical facts established as a result of this research fit perfectly into the following scheme:Carbon monoxide oxidation catalysts can be in two states: (1) the surface of catalyst clusters is covered with a layer of adsorbed carbon monoxide molecules; (2) the surface of catalyst clusters is covered with a layer of adsorbed oxygen molecules;The reaction mechanisms in these two states of the catalyst differ sharply from each other: in state (1), oxygen from the gas phase interacts with adsorbed carbon monoxide; in state (2), carbon monoxide from the gas phase interacts with adsorbed oxygen; the rate of reactions in state (1) is 4–5 times lower than in state (2);The activation energy of the carbon monoxide oxidation reaction in state (1) does not depend on the oxygen concentration; however, the pre-exponential factor is proportional to the oxygen concentration. Moreover, when the catalyst powder is diluted with inert alumina powder, the activation energy does not change, and the pre-exponential factor decreases in proportion to the degree of catalyst dilution;The transition of the catalyst from state (1) to state (2) occurs abruptly with a decrease in the concentration of carbon monoxide in the gas phase below a certain limit, depending on the type of catalyst; with a decrease in the concentration of carbon monoxide in the gas phase, the avalanche-like clearing of catalyst surface from a layer of adsorbed carbon monoxide molecules takes place;The transition from state (1) to state (2) is reversible; the reverse transition from state (2) to state (1) occurs with some delay as the concentration of carbon monoxide increases;The carbon monoxide oxidation reaction in both states of the catalyst obeys the first-order kinetic law, and the “minus first-order” known from the literature for this reaction is associated with some efficient processes arising from the superposition of processes (1) and (2).

To analyze the behavior of the gas-sensitive material of a semiconductor carbon monoxide sensor, from our point of view, the most important thing is the hysteresis during temperature cycling. Even in the presence of carbon monoxide, palladium, which decorates tin oxide of the gas-sensitive layer, passes into a highly active state, when the temperature rises above the critical one. With a sharp decrease in temperature below the critical value, this highly active (oxidized) state of the catalyst is preserved, and palladium oxide can, for some time, be a reservoir of oxygen consumed for the oxidation of carbon monoxide. Firstly, in this case, the rate of oxygen consumption is proportional to the concentration of carbon monoxide in the air. Secondly, the integral oxygen consumption is proportional to the time until all oxygen is consumed.

Thus, with a cyclic change in temperature in the high-temperature region, palladium is oxidized. It passes into a highly active state, and with a sharp decrease in temperature, the oxygen stored in the oxide is consumed for the oxidation of carbon monoxide. At the same time, tin oxide, which forms the basis of the sensitive layer, helps to transform these changes into an electrical signal: when palladium is oxidized, p-n junctions occur at the contacts of palladium oxide and tin oxide (since tin dioxide is an n-type semiconductor and palladium oxide is a p-type semiconductor), disappearing as oxygen is consumed from palladium oxide in the carbon monoxide oxidation reaction. This process of appearance and disappearance of p-n junctions leads to a corresponding change in the concentration of charge carriers in tin dioxide and, consequently, to a change in its electrical conductivity, which is what we observe in experiments.

The analysis of the sensor response and an attempt to restore the mechanism and the kinetics of the process using these data is a typical solution of reverse kinetic problem. It is rather difficult to solve this problem, which generally does not guarantee an unambiguous solution. A more complicated problem is proving the accuracy of the model. As a rule, this could be conducted by the independent measurement of rate constants of the reactions used in the kinetic model and by the comparison of the rate constants obtained from the model and from direct measurements. Another very important approach to the proof of the model of the heterogeneous process is the investigation of the surface state.

This last approach was used, for example, in the paper [32]. The authors of this paper investigated the roughness of the surface of a Pd single crystal in two states: after oxidation and after adsorption of CO. It was shown that the roughness of the surface is considerably different, and it is possible to distinguish between these two states of the surface.

It was shown that the oxidized (rough) state of the palladium surface leads to a faster reaction of CO oxidation compared to a slower reaction on a reduced (smooth) surface. The transition between these two states of the surface is reversible, but the authors observed hysteresis: when they increased the concentration of CO, the reaction rate followed the slower branch of the curve. A subsequent decrease in concentration leads to the reaction by this slower branch, but at a certain concentration, the reaction rate increases suddenly, and the reaction follows the faster branch of the kinetic curve. The difference between these two states was clear not only from kinetic measurements but also from the measurement of the surface roughness.

Similar results were observed by us from kinetic measurements. In addition to the measurements and solution of the reverse kinetic problem with curves analogous to Figure 4, we performed several independent experiments reported in [36]. For kinetic experiments, we traditionally measured the concentration of carbon monoxide in the outlet of the reactor as a function of the residence time of the gas in the reactor. This reaction is of the first kinetic order, that is, a drop of CO concentration is described by the equation d[CO]/d*t* = −*k*[CO], where [CO] is CO concentration and *k* is the reaction rate constant. The reaction rate constants and activation energies calculated from these plots coincide with the values obtained from the simulation of the curves similar to those presented in Figure 4. Another experiment was performed to demonstrate that, in fact, there is a certain concentration of CO that leads to a sudden increase in reaction rate when CO concentration is below this threshold. For this, we measured the concentration of CO in the outlet of the reactor as a function of the inlet concentration. Really, below a certain concentration in the inlet, concentration in the outlet decreases as a step by a factor of 4–5, which means a stepwise increase in reaction rate.

Results of our experiments (both kinetics and sensor responses) together with analyses of publications confirm, therefore, our model of the sensor response to CO.

## 4. Discussion

To analyze gases with low-selective sensors under non-stationary temperature conditions, it is necessary to choose an optimal algorithm for processing multidimensional data arrays. The best way to interpret such data is to find a functional relationship between the electrical conductivity of the sensor and the time after the start of the cycle (or the time-dependent temperature of the sensor). To find this dependence, it is necessary to understand the mechanism of the process of chemisorption of analyte gases and their catalytic interaction with chemisorbed oxygen. To find the mechanism of the process and describe its kinetics, a catalytic experiment was carried out, including the determination of the composition of carbon monoxide oxidation products at various temperatures of the gas-sensitive material.

The results of the catalytic experiment showed that palladium clusters can be in two states, depending on the prehistory of the process. At a high temperature and a low concentration of carbon monoxide in the air, palladium is in an oxidized state, while at its high concentration and a lower temperature of about 100–200 °C, it turns out to be covered with a layer of chemisorbed carbon monoxide molecules. In this case, the rate of carbon monoxide oxidation on the catalyst surface of palladium clusters in the oxidized state is about 4–5 times higher than the rate of oxidation in the state with chemisorbed carbon monoxide. An important feature of this process is the sharp transition from one state to another, even under conditions close to isothermal. This feature leads to the fact that with a sharp decrease in the catalyst temperature, its oxidized state can “freeze”, that is, this state will remain even in the presence of carbon monoxide at relatively low temperatures of about 100 °C.

Used as a sensor material, tin dioxide is an n-type semiconductor, and palladium oxide is a p-type semiconductor. The contact of a tin dioxide particle with a cluster of palladium oxide leads to the depletion of tin dioxide in charge carriers (electrons) and a decrease in its electrical conductivity.

Taking into account the results on the kinetics of carbon monoxide oxidation on the surface of tin dioxide decorated with palladium clusters, we can propose the following chemical mechanism for the operation of a semiconductor sensor in a pulsed mode: When the gas-sensitive layer is heated rapidly in the air, palladium is oxidized to palladium oxide. Then, upon rapid cooling to a temperature of the order of 100 °C, this oxidized state “freezes”, and at this temperature, palladium oxide relatively slowly, over a time of the order of several seconds, interacts with carbon monoxide molecules, oxidizing them to carbon dioxide. In this case, palladium oxide itself is reduced, this leads to a gradual disappearance of the p-n junction, a decrease in the depletion of tin dioxide in electrons, and an increase in the conductivity of the gas-sensitive layer. In this case, due to the fact that the reaction has the first kinetic order with respect to carbon monoxide, an increase in the conductivity of tin dioxide will be linear in time in the initial section, and this increase depends almost linearly on the concentration of carbon monoxide in the air.

The results of kinetic experiments helped us to find a functional dependence relating the electrical resistance of the SnO_2_–PdO_x_ sensor to its temperature in a carbon monoxide environment and use this functional dependence to perform a selective analysis of the gaseous medium.

The qualitative analysis algorithm can be based on estimating the slope of the dependence of the electrical conductivity of the SnO_2_–PdO_x_ sensor on time for the last third of the measurement cycle (from 10 to 15 s from the beginning of the cycle, Table 1).
*R*_1_ = lg(σ(τ =15)/σ(τ = 10)).(4)

As can be seen from Table 1, if the decimal logarithm of the ratio for the corresponding electrical conductivities of the sensor *R*_1_ exceeds 0.15, then the detected analyte gas is carbon monoxide, and if this value is less than 0.15, then the detected analyte gas is hydrogen.

For quantitative analysis, we can use a calibration curve (Figure 5) built from the values of the electrical conductivity of the sensor at the end of the measurement cycle at τ = 14.5 s.

A comparative analysis of calibration curves 1 and 2 in Figure 5 shows that the dependences of the electrical conductivity of the SnO_2_–PdO_x_ sensor at the end of the measurement cycle on the concentration of analyte gases are close to linear. At the same time, the deviations from the calibration data for carbon monoxide and hydrogen in air at concentrations from 0 to 100 ppm are not statistically significant. This allows us to conclude that when carrying out a quantitative analysis of carbon monoxide and hydrogen in the air using the SnO_2_–PdO_x_ sensor, it is quite possible to use a general linear calibration dependence.

The results of the kinetic experiments can also be used for the interpretation of the measuring scheme of the sensor TGS2442 (Figaro Inc.; see, for example, [36]). The recommendation of the company consists of heating the sensing layer for 14 ms up to a high temperature, followed by switching off for 986 ms. The duration of the whole measuring cycle is, therefore, 1 s. For the determination of carbon monoxide concentration, the resistance of the sensing layer is measured during the last 10 ms of the measurement cycle. 

The chemical process responsible for the measurement is the same as described above. During the sharp heating at the beginning of the cycle, the Pd decoration of the sensing layer is oxidized, and after switching the heater off, this palladium oxide serves as a reservoir of oxygen for the oxidation of CO. The last 10 ms of the measurement cycle enable the determination of integral oxygen consumption in CO oxidation and, therefore, the determination of current CO concentration because the duration of the process is constant and equals 1 s. The conductivity of the sensing layer is controlled by the formation of a p–n junction between n-type SnO_2_ and p-type PdOx during sharp heating and the disappearance of this junction due to the reduction of PdO in the reaction with CO in the low-temperature phase of the cycle. However, an uncontrolled heating and cooling process can lead to errors in CO measurement.

## 5. Conclusions

An effective method for increasing the selectivity of the analysis of gaseous media by metal oxide sensors is the use of non-stationary temperature regimes capable of separating in time the processes of chemisorption, the chemical interaction of the target gas with chemisorbed oxygen, and the desorption of the resulting reaction products. The dependences of the electrical conductivity of the sensor on time (or temperature) obtained in such experiments have differences that are specific for each analyte gas. These differences between the curves help to perform a selective analysis of gaseous media, but this raises the problem of optimizing algorithms for processing multidimensional data arrays in terms of the resources required for their implementation.

The most efficient algorithms for processing multidimensional data can be obtained by analytically solving equations that relate the electrical conductivity of a sensor to its temperature. However, obtaining such equations requires an understanding of the mechanism and kinetics of the processes occurring on the sensor’s surface. In our work, a special kinetic experiment was carried out, which made it possible to reveal the features of the mechanism of carbon monoxide oxidation on the surface of the SnO_2_–PdO_x_ sensor. The experiment showed that with a sharp decrease in temperature below the critical value, the highly active (oxidized) state of palladium oxide is preserved, so it can be a reservoir of oxygen consumed for the oxidation of carbon monoxide.

Experiments were carried out to determine the response of the SnO_2_–PdO_x_ sensor in non-stationary temperature conditions in gaseous media containing carbon monoxide and hydrogen at different concentrations. As a result of these experiments, we revealed the features of the change in electrical conductivity during cooling of the SnO_2_–PdO_x_ sensor in a carbon monoxide environment, which allowed us to propose a simple and effective algorithm for the selective analysis of conditionally one-component gas systems.

## Figures and Tables

**Figure 1 sensors-23-03730-f001:**
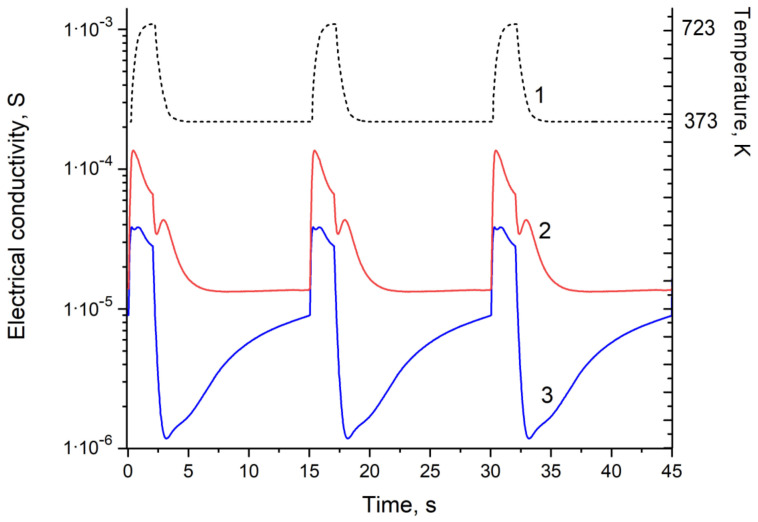
Change in temperature (1) and electrical conductivity of the MOX sensor at 100 ppm hydrogen (2) and 100 ppm carbon monoxide (3) in the air over three measurement cycles.

**Figure 2 sensors-23-03730-f002:**
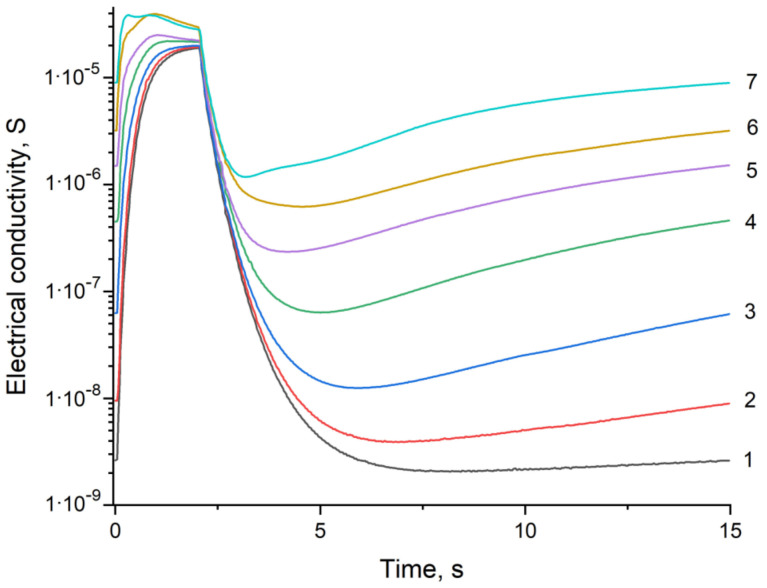
Change in electrical conductivity of the SnO_2_–PdO_x_ sensor for carbon monoxide concentrations in air in a range from 0 to 100 ppm (curves 1–7) during 1 measurement cycle.

**Figure 3 sensors-23-03730-f003:**
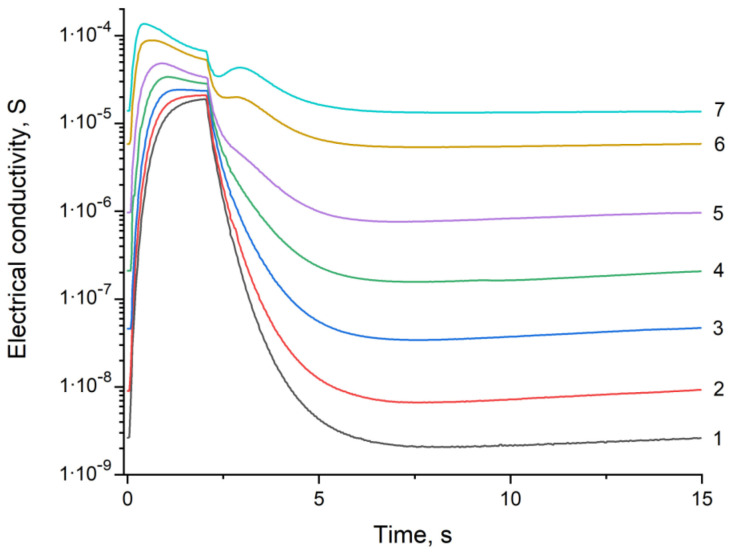
Change in electrical conductivity of the SnO_2_–PdO_x_ sensor for hydrogen concentrations in air from 0 to 100 ppm (curves 1–7) during 1 measurement cycle.

**Figure 4 sensors-23-03730-f004:**
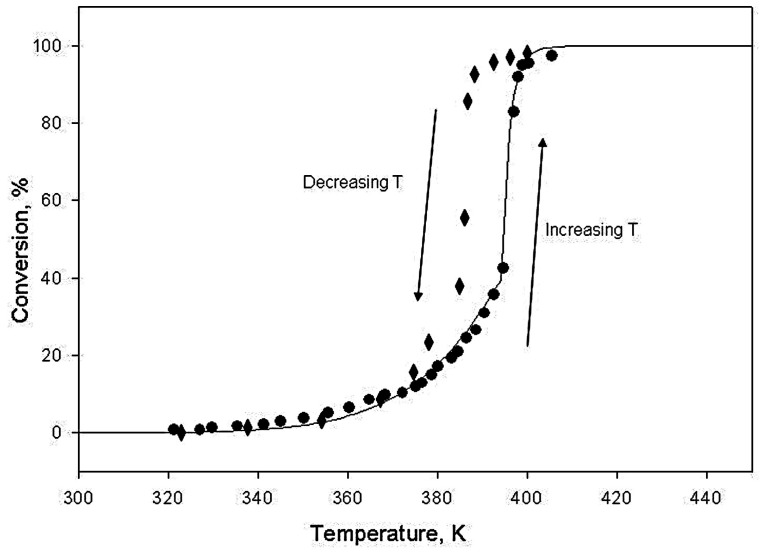
Hysteresis of the dependence of the degree of carbon monoxide conversion on temperature for the catalyst SnO_2_–PdO_x_. Circles correspond to increasing temperature, rhombs—decreasing temperature, respectively.

**Figure 5 sensors-23-03730-f005:**
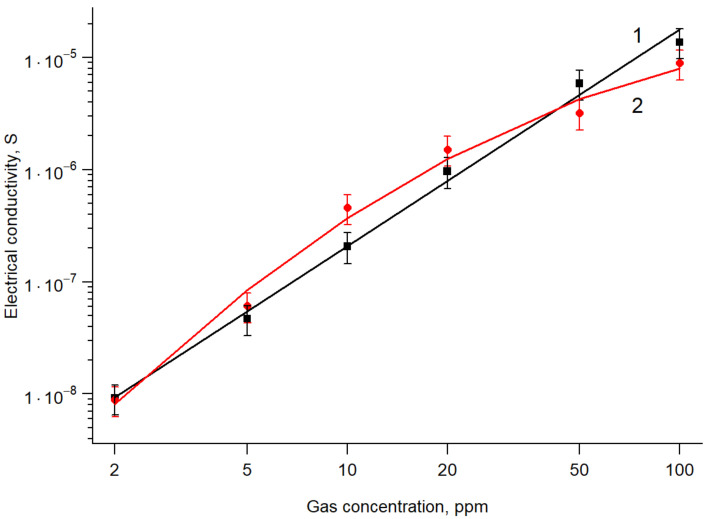
Calibration curves of the SnO_2_–PdO_x_ sensor when determining hydrogen (curve 1) and carbon monoxide (curve 2).

**Table 1 sensors-23-03730-t001:** The decimal logarithm of the ratio of electrical conductivities *R*_1_ of the SnO_2_–PdO_x_ sensor, at the beginning and at the end of the last third of the measurement cycle for hydrogen and carbon monoxide in the air.

Analyte Concentration, ppm	*R*_1_ Value for H_2_	*R*_1_ Value for CO
0	0.080	0.080
2	0.108	0.248
5	0.099	0.383
10	0.103	0.367
20	0.066	0.284
50	0.029	0.255
100	0.009	0.192

## Data Availability

Details about the kinetics of the CO oxidation over noble metal catalysts can be found in our publication [35].

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
