# Peer review of "Kinetics of Chemisorption on the Surface of Nanodispersed SnO2–PdOx and Selective Determination of CO and H2 in Air"

_sensors, 2023, doi:10.3390/s23073730_

Round 1
Reviewer 1 Report
This is an interesting paper, which I enjoyed reading. It studies the kinetic effects of temperature changes in catalyst-loaded tin oxide materials in the presence of hydrogen or carbon monoxide. In the paper there is, generally, a thorough discussion of results in a plausible manner, according to the experiments available. I have a short set of recommendations that authors should try to address in a revised version of their paper. These are as follows:
1. To some extent the paper is speculative, as the mechanisms of response indicated have not been observed directly. Sometimes it is difficult to completely validate a model using only gas response measurements (i.e., conductivity changes). Maybe the use of some spectroscopic methods should be envisaged here to better identify surface species or the changes in the oxidation states of the catalyst. In any case I think that these deserves to be discussed in the revised paper.
2. Figure 1 shows that the response features upon a sudden temperature change of the sensor differ for carbon monoxide and carbon dioxide. These changes (and underlaying reasons) are not discussed in detail in the manuscript, as no mention seems to appear in the original manuscript.
3. You indicate that the response is the ration between the sensor conductivity while exposed to the target species to the conductivity in air (at 14.5 s). Why this value in particular was retained? How the full vector of data available is used, to exploit the differences in response features? This deserves to be clarified.
4. So far, the study has been conducted for single gaseous species. How this approach could be extended further to gas mixtures?
5. Figure 5 shows the calibration curves obtained for hydrogen and carbon monoxide. Can you please add error bars so the uncertainty associated to your methods is accounted for? Also given the closeness between the two curves, is it really possible to identify these two species with a single sensor?
6. Sensory response is used throughout the manuscript and I think this should be changed to ‘sensor response’. Sensory is mostly used in the context of the organoleptic assessment of, for example, foodstuffs and beverages via test panels.
Author Response
Reviewer 1:
This is an interesting paper, which I enjoyed reading. It studies the kinetic effects of temperature changes in catalyst-loaded tin oxide materials in the presence of hydrogen or carbon monoxide. In the paper there is, generally, a thorough discussion of results in a plausible manner, according to the experiments available. I have a short set of recommendations that authors should try to address in a revised version of their paper. These are as follows:
Acknowledgment.The authors are sincerely grateful to the Reviewer 1 for comments and hope that the revised version of the article will contribute to a better understanding of the results obtained.
Comment 1.1.To some extent the paper is speculative, as the mechanisms of response indicated have not been observed directly. Sometimes it is difficult to completely validate a model using only gas response measurements (i.e., conductivity changes). Maybe the use of some spectroscopic methods should be envisaged here to better identify surface species or the changes in the oxidation states of the catalyst. In any case I think that these deserves to be discussed in the revised paper.
Response 1.1.The analysis of sensor response and an attempt to restore the mechanism and the kinetics of the process using these data is a typical case of solution of reverse kinetic problem. It is rather difficult to solve this problem, which generally does not guarantee an unambiguous solution. More complicated problem is the proof of the accuracy of the model. As a rule, this could be done by the independent measurement of rate constants of the reactions used in the kinetic model and by the comparison of the rate constants obtained from the model and from direct measurements. Another, very important approach to the proof of the model of heterogeneous process is the investigation of the surface state.
This last approach was used, for example, in the paper [33]. The authors of this paper investigated the roughness of the surface of Pd single crystal in two states: after oxidation and after adsorption of CO. It was shown, that the roughness of the surface is rather different, and it is possible to distinguish between these two states of the surface.
It was shown that the oxidized (rough) state of the palladium surface leads to the faster reaction of CO oxidation compared to slower reaction on reduced (smooth) surface. The transition between these two states of the surface is reversible, but the authors observed hysteresis: when they increased the concentration of CO, the reaction rate follows the slower branch of the curve. Subsequent decrease in concentration leads the reaction by this slower branch, but at certain concentration the reaction rate increases suddenly, and the reaction follows the faster branch of the kinetic curve. The difference between these two states was clear not only from kinetic measurements, but also from the measurement of the surface roughness.
Similar results were observed by us from kinetic measurements. In addition to the measurements and solution of reverse kinetic problem with curves analogous to the Fig. 4, we performed several independent experiments reported in [36]. We measured traditionally for kinetic experiments concentration of carbon monoxide in the outlet of the reactor as a function of residence time of gas in the reactor. This reaction is of the first kinetic order, that is drop of CO concentration is described by the equation d[CO]/dt = −k[CO], where [CO] is CO concentration, k – reaction rate constant. The reaction rate constants and activation energies calculated from these plots coincide with the values obtained from the simulation of the curves similar to those presented in Fig. 4. Another experiment was performed to demonstrate that in fact there is certain concentration of CO leading to sudden increase in reaction rate, when CO concentration if below this threshold. For this, we measured the concentration of CO in the outlet of the reactor as a function of inlet concentration. Really, below certain concentration in the inlet, concentration in the outlet decreases as a step by a factor of 4–5, this means a stepwise increase in reaction rate.
Results of our experiments (both kinetics and sensor responses) together with analysis of publications confirm, therefore, our model of the sensor response to CO.
The required changes have been made to the article, lines 330-366.
Comment 1.2. Figure 1 shows that the response features upon a sudden temperature change of the sensor differ for carbon monoxide and carbon dioxide. These changes (and underlaying reasons) are not discussed in detail in the manuscript, as no mention seems to appear in the original manuscript.
Response 1.2.As shown in Figure 1, changing the temperature of the sensor leads to a significant change in the electrical conductivity of the gas-sensitive layer. In the first section (2 seconds), the sensor is heated from a temperature of 373 K to a temperature of 723 K. In this section, the electrical conductivity of the gas-sensitive layer increases due to an increase in the concentration of charge carriers in the semiconductor. In the second section (13 seconds), the sensor cools down from a temperature of 723 K to a temperature of 373 K. The general trend is aimed at reducing the electrical conductivity of the gas-sensitive layer. However, when determining hydrogen at the initial stage of sensor cooling, one can observe an additional extremum, which is characteristic of hydrogen. The nature of this extremum is apparently associated with the ability of hydrogen molecules to dissociate into atoms and oxidize to cations that can make an additional contribution to the electrical conductivity of the gas-sensitive layer. When determining carbon monoxide in the cooling section, one can observe a minimum of electrical conductivity, after which its increase follows. This increase in electrical conductivity is a specific feature of carbon monoxide and is associated with a change in the mechanism of CO interaction with the palladium surface.
The required changes have been made to the article, lines 148-163.
Comment 1.3.You indicate that the response is the ration between the sensor conductivity while exposed to the target species to the conductivity in air (at 14.5 s). Why this value in particular was retained? How the full vector of data available is used, to exploit the differences in response features? This deserves to be clarified.
Response 1.3.One of the objectives of this work is to develop algorithms for a simple and compact gas analyzer suitable for autonomous selective analysis of a gaseous medium in hard-to-reach places. Control and data processing in such gas analyzers is implemented using relatively simple and low-performance microcontrollers, instead of high-performance processors.
The authors of the article are familiar with the multivariate calibration methods, which provide some increase in the accuracy of quantitative analysis due to a significant increase in the computational complexity of the algorithms, however, the main task of our work was different – to develop the principles of qualitative analysis of the gaseous medium that do not require the use of significant computing resources.
Comment 1.4.So far, the study has been conducted for single gaseous species. How this approach could be extended further to gas mixtures?
Response 1.4.There are several problems of selective analysis. The first of these is the qualitative and quantitative analysis of one-component gas systems, such as “carbon monoxide in air” and “hydrogen in air”. Determination of the composition of gas mixtures in a continuous range of concentrations is a more complex problem, the solution of which is beyond the scope of this work.
Comment 1.5.Figure 5 shows the calibration curves obtained for hydrogen and carbon monoxide. Can you please add error bars so the uncertainty associated to your methods is accounted for? Also given the closeness between the two curves, is it really possible to identify these two species with a single sensor?
Response 1.5.Error bars are added to the calibration lines shown in Figure 5, which are used not for qualitative, but for quantitative analysis of the gaseous medium. In order to distinguish carbon monoxide from hydrogen, the data presented in Table 1 is used.
As can be seen from Table 1, if the decimal logarithm of the ratio for the corresponding electrical conductivities of the sensor R1 exceeds 0.15, then the detected analyte gas is carbon monoxide, and if this value is less than 0.15, then the detected analyte gas is hydrogen.
Comment 1.6.Sensory response is used throughout the manuscript and I think this should be changed to ‘sensor response’. Sensory is mostly used in the context of the organoleptic assessment of, for example, foodstuffs and beverages via test panels.
Response 1.6.The required changes have been made to the article.

Reviewer 2 Report
The authors present a sensor based on SnO2-PdO nano particle-based metal oxide semiconductor platform to study the improvement of selective and quantitative sensing of CO and H2 in air. The authors employ the use of kinetic analysis at non stationary temperature regimes to obtain improved selectivity and quantitative measurements. The paper is well structured, the experimental process is sound and the use of English is good. The paper can be considered for publication in Sensors after the following issues have been addressed:
1. A schematic diagram of the sensor device structure as well as the gas analysis/measurement setup should be added to the manuscript.
2. The material characterization data of SnO2 and PdO nano particles such as XRD, XPS, TEM etc., should be included as a figure to provide useful information in terms of the materials aspect of the sensor fabrication. It seems that the authors are just referring the readers to look at reference 30 (the authors’ previous paper) for these characterization results which may not be appropriate for the originality requirements of Sensors Journal on results and methodology. The authors must address this and may consult with the editors on the policy of reporting results from a previous publication to this journal.
3. The qualitative analysis algorithm based on R1 values generated by each type of gas analyte is impressive. However these were generated for one species of analyte at a time. It would make the paper complete if the authors actually used this analysis algorithm to actually determine the presence of both H2 and CO in mixed state and demonstrate both qualitative and quantitative readings when both gases are present.
After these issues have been addressed by the authors, the paper can be considered for publication in Sensors.
Author Response
Reviewer 4
The authors present a sensor based on SnO2-PdO nano particle-based metal oxide semiconductor platform to study the improvement of selective and quantitative sensing of CO and H2 in air. The authors employ the use of kinetic analysis at non stationary temperature regimes to obtain improved selectivity and quantitative measurements. The paper is well structured, the experimental process is sound and the use of English is good. The paper can be considered for publication in Sensors after the following issues have been addressed.
Acknowledgment.The authors are sincerely grateful to the Reviewer 4 for comments and hope that the revised version of the article will contribute to a better understanding of the results obtained.
We are very sorry, unfortunately, the review was recieved much later, then the reviews of other reviewers, and we were not able to meet it, because the decision about "major revisions" was obtaind before.
- A schematic diagram of the sensor device structure as well as the gas analysis/measurement setup should be added to the manuscript.
The scheme of the thick film gas sensor used in our experiments was presented several times in our publication. The sensor consists of thin dielectric substrate with dimension of about 2.5x0.5 mm, the total thickness of the sensor is of about 0.1 mm. One side of the substrate is covered by platinum based resistive material (a composite, containing Pt nanoparticles, Al2O3 and glass binder), another side is equiped with Pt contacts to the sensing layer, the contacts are covered by thick-film sensing layer. The sensor chip is suspended in TO8 holder using 20 micron in diameter Pt wires.
We are not sure that it is reassonable to repeat this picture once more, however, if the reviewer insists to this, we are ready to consider this possibility.
2. The material characterization data of SnO2 and PdO nano particles such as XRD, XPS, TEM etc., should be included as a figure to provide useful information in terms of the materials aspect of the sensor fabrication. It seems that the authors are just referring the readers to look at reference 30 (the authors’ previous paper) for these characterization results which may not be appropriate for the originality requirements of Sensors Journal on results and methodology. The authors must address this and may consult with the editors on the policy of reporting results from a previous publication to this journal.
Thank you very much for the comment. These results were, in fact, published in our previous paper (ref. 30). If the reviewer insists in this, we can repeat the pictures in this paper. However, may be, it is sufficient to give a reference? We are not sure that an increase of paper size is very useful.
3. The qualitative analysis algorithm based on R1 values generated by each type of gas analyte is impressive. However these were generated for one species of analyte at a time. It would make the paper complete if the authors actually used this analysis algorithm to actually determine the presence of both H2 and CO in mixed state and demonstrate both qualitative and quantitative readings when both gases are present.
We considered the possibility of simultaneous detection of CO and H2 by single sensor. For this, the algorithm should be different from this one, described in our paper submitted now. The description of this algorithm can be found in our paper A.V.Shaposhnik, P.V.Moskalev, K.L.Chegereva, A.A.Zviagin, A.A.Vasiliev. Selective gas detection of H2 and CO by a single MOX-sensor. Sensors and Actuators B: Chemical. Volume 334, 1 May 2021, 129376.
Reviewer 3 Report
The present manuscript deals with an interesting topic, i.e. the kinetics of chemisorption on nano-SnO2-PdOx material and the selective sensing of CO and H2 in air. The manuscript should be published upon minor revisions. There are some issues to be improved:
· In the introduction section, some more recent papers concerning some possible explanation of the chemiresistive mechanism should be cited such as: 10.1016/j.electacta.2020.137611, 10.1039/c7ta09535j, 10.1021/acsami.1c08236 and 10.3390/nano12152696;
· authors state that the sensor response was computed considering the electrical conductivity at 14.5 s from the beginning of the measurement. This is an uncommon procedure, therefore they should explain the reason why a timeframe of 14.5 s was considered and report some literature to corroborate this point;
· calibration curve reported in Figure 5 doesn’t have scientific meaning. Each point should be reported with the relative error bar and, in addition, a linear regression fitting should have been done
· within the whole manuscript, several mistakes are present. Therefore, the paper should be revised accordingly.
Author Response
Reviewer 2:
The present manuscript deals with an interesting topic, i.e. the kinetics of chemisorption on nano-SnO2-PdOx material and the selective sensing of CO and H2 in air. The manuscript should be published upon minor revisions. There are some issues to be improved:
Acknowledgment.The authors are sincerely grateful to the Reviewer 2 for comments and hope that the revised version of the article will contribute to a better understanding of the results obtained.
Comment 2.1.In the introduction section, some more recent papers concerning some possible explanation of the chemiresistive mechanism should be cited such as: 10.1016/j.electacta.2020.137611, 10.1039/c7ta09535j, 10.1021/acsami.1c08236 and 10.3390/nano12152696.
Response 2.1.The required changes have been made to the article, lines 65-66, 553-564.
Comment 2.2.Authors state that the sensor response was computed considering the electrical conductivity at 14.5 s from the beginning of the measurement. This is an uncommon procedure, therefore they should explain the reason why a timeframe of 14.5 s was considered and report some literature to corroborate this point.
Response 2.2.The choice of a time point of 14.5 seconds from the beginning of the measurement cycle was due to the maximum difference between the values of the electrical conductivity of the sensor obtained at different concentrations of analyte gases.The required changes have been made to the article, lines 176-179.
Comment 2.3.Calibration curve reported in Figure 5 doesn’t have scientific meaning. Each point should be reported with the relative error bar and, in addition, a linear regression fitting should have been done.
Response 2.3.Error bars and regression lines have been added in Figure 5.
Comment 2.4.Within the whole manuscript, several mistakes are present. Therefore, the paper should be revised accordingly.
Response 2.4.The paper has been revised more accurately.
Reviewer 4 Report
The topic and content of the paper are quite interesting to readers. Some question / suggestions are as follows:
1. Some typos need to be corrected. For example, the “thesenssory” at line 54, “shave” at line 114, “Conductivityin” at line 218 etc.
2. It is difficult to understand What the “first-order reaction” at line 214 mean.
3. For the first paragraph of Section 3.2, “When studying the process of carbon monoxide oxidation on the surface of a palladium-containing catalyst, it was shown that an abrupt increase in this process is not of a temperature, but of a concentration nature”, this is also difficult to understand. It will be better to have a figure, or explain this sentence combined with some figures before.
4. What’s the relation between the “40 Hz” at line 146 and “38.3 Hz” at line 163?
Author Response
Reviewer 3:
The topic and content of the paper are quite interesting to readers. Some question / suggestions are as follows.
Acknowledgment.The authors are sincerely grateful to the Reviewer 3 for comments and hope that the revised version of the article will contribute to a better understanding of the results obtained.
Comment 3.1.Some typos need to be corrected. For example, the “thesenssory” at line 54, “shave” at line 114, “Conductivityin” at line 218 etc.
Response 3.1.The typos have been corrected.
Comment 3.2.It is difficult to understand What the “first-order reaction” at line 214 mean.
Response 3.2.The reaction of CO oxidation follows the first order kinetic equation, that is
d[CO]/dt = − k[CO],
where [CO] is the concentration of CO, k – reaction rate constant.
Comment 3.3.For the first paragraph of Section 3.2, “When studying the process of carbon monoxide oxidation on the surface of a palladium-containing catalyst, it was shown that an abrupt increase in this process is not of a temperature, but of a concentration nature”, this is also difficult to understand. It will be better to have a figure, or explain this sentence combined with some figures before.
Response 3.3.These results were published in detail in [A.A. Vasiliev, A.S. Lagutin, and Sh.Sh. Nabiev. Optimization of CO Oxidation Catalysts for Thermocatalytic and Semiconducting Gas Sensors. Russian Journal of Inorganic Chemistry, 2020, Vol. 65, No. 12, pp. 1948–1957], Fig. 2. The paper is attached.
If we set the temperature and the concentration of the temperature of the reactor near and a little below the abrupt increase in CO conversion, at temperature of 382.5 C (Fig. 4) and start to decrease slowly the concentration of CO in the inlet of the reactor, at certain concentration of CO the concentration in the outlet of the reactor decreases as a step. This means a stepwise increase in the reaction rate due not to an increase in temperature and to the heat exchange crisis, but to a decrease in CO concentration and, therefore, due to a change in the reaction mechanism. At high CO concentration the surface of the catalyst is poisoned by adsorbed CO, but when CO concentration decreases below certain threshold, the poisoning suddenly disappears.
Comment 3.4.What’s the relation between the “40 Hz” at line 146 and “38.3 Hz” at line 163?
Response 3.4.The typos in lines 145, 175 have been corrected.

Round 2
Reviewer 2 Report
The authors have addressed the issues brought up by this reviewer and have satisfactorily revised their manuscript accordingly. This manuscript is ready for publication in Sensors.